## Research Article

depression; mobile health; low income countries; mental health; digital health

**Corresponding author:**
Bisola A. Olayemi;
Email: olayemibisola0@gmail.com

# Facilitating use of mobile health in mental health care: Perspectives of patient, caregiver and provider in Nigeria

Bisola A. Olayemi[1] , Tobi Fatodu[1], Adeyinka Adefolarin[2], Oluwatoyin Olujimi[1], John C. Fortney[3], Oye Gureje[1] and Lola Kola[1,4]

[1]WHO Collaborating Centre for Research and Training in Mental Health, Neurosciences and Drug and Alcohol abuse. Department of Psychiatry, College of Medicine, University of Ibadan, Nigeria; [2]Department of Health promotion and Education, Faculty of Public Health, University of Ibadan, Nigeria; [3]Department of Psychiatry, School of Medicine, University of Washington, Seattle, WA, USA. VA Health Systems Research, Center of Innovation for Veteran-Centered and Value-Driven Care, VA Puget Sound, Seattle, WA and [4]King's College London Institute of Psychiatry Psychology & Neuroscience, London, UK

## Abstract

Mobile health (mHealth) interventions offer promising ways to enhance access and continuity of mental health services in low-resource settings. However, little is known about the perspective of end users in routine primary care in Nigeria regarding the role of mHealth in mental health care. This qualitative study explored the perspectives of patients, caregivers and healthcare providers on the use of mHealth tools to support access to and continuity of mental health care in Nigeria. Seventeen participants, including persons with lived experience of depression (n=7), caregivers (n=3), and primary healthcare workers (n=7), were purposively recruited from nine primary health clinics in Ibadan. Interviews were conducted in Yoruba, transcribed, translated into English, and analysed inductively using NVivo 15. Participants identified phone calls, Short Message Service (SMS) reminders, and audiovisual content as key facilitators of engagement, self-care and adherence. Caregivers valued direct communication with providers, while healthcare workers used mobile tools for reminders, follow-up and patient education. Flexible use of next-of-kin contacts helped overcome digital barriers. The findings demonstrate that user-friendly mHealth tools are feasible for supporting mental health care in Nigeria, but their success depends on coupling technology with human-centred communication to ensure equitable and continuous care.

## Impact statement

This formative qualitative study provides early-stage evidence on the use of mobile health interventions such as SMS, phone calls and educational videos in enhancing access to and continuity of mental health care in low-resource settings. Exploring the perspectives of patients, caregivers and healthcare providers, it highlights the value of use of digital solutions in addressing Nigeria's large mental health treatment gap. The findings inform scalable strategies for implementing equitable digital mental health interventions and offer practical guidance for policymakers and practitioners seeking to integrate mobile health tools into primary mental health systems in Nigeria and other low- and middle-income countries.

## Introduction

Equitable access to high-quality mental health care remains a great challenge globally, particularly in low- and middle-income countries (LMICs) where under-resourced health systems often leave individuals living with mental health conditions without the care and support they need (Wainberg et al., 2017; Patel et al., 2018). In Nigeria, this challenge is especially acute, with an estimated 80% of individuals experiencing mental health conditions receiving no formal care (Fadele et al., 2024). This treatment gap is attributed to a combination of structural and personnel factors, including a critical shortage of trained mental health professionals, poor infrastructure, limited funding, poor knowledge and significant stigma surrounding mental illness (Kakuma et al., 2011). Consequently, many affected individuals are left to cope without access to appropriate, timely, or culturally sensitive mental health services. Addressing these barriers requires innovative strategies that can extend access to improved mental health care.

A promising avenue to address these gaps is the use of digital health technologies. To improve access to mental health care, the WHO strongly recommends the widespread use of technology as a possible solution for transforming services (World Health Organization, 2019;

Okoro et al., 2024). Digital health tools, especially mobile health (mHealth) interventions, offer new opportunities to strengthen mental health systems by addressing access, communication and service delivery gaps. These tools are increasingly becoming an important channel for health system strengthening and patient engagement (Erku et al., 2023). In Nigeria, mobile phone penetration has grown rapidly in recent years as an essential tool in daily life across age groups. The country reports a teledensity rate of 97.9%, reflecting near-universal access to mobile phones (First Fiduciary, 2022). Recent studies have documented increasing uptake of mHealth applications in Nigeria for various health domains, demonstrating acceptability and feasibility within the local context (Kola et al., 2025). Emerging evidence also suggests growing interest in using mHealth for mental health support, with mobile platforms facilitating psychoeducation, appointment reminders and symptom tracking (Dominiak et al., 2024). These developments underscore the transformative potential of mobile technologies to expand the reach and equity of mental health services in Nigeria.

While emerging evidence highlights the potential benefits of mHealth in supporting patient self-care, such as facilitating access to psychoeducational resources, much of this research comes from high-income countries. In contrast, little is known about how healthcare users in Nigeria and other LMICs engage with these tools in the context of their daily lives (Fortuna et al., 2022). In addition to supporting self-care, mHealth tools also offer opportunities to enhance communication and collaboration between patients and mental health care providers (Fitzpatrick, 2023). In LMICs context, mHealth is primarily being used to overcome significant barriers to traditional care, such as stigma, geographical distance and a shortage of specialists (Carter et al., 2021). Digital platforms such as telemedicine and mobile messaging can facilitate real-time information sharing, reduce missed appointments and ensure more consistent follow-up care. For example, SMS reminders have been shown to improve appointment adherence among individuals newly diagnosed with depression in Nigeria (Fashoto et al., 2025). However, several challenges may limit the integration of mHealth into routine clinical practice, including provider burden and limited digital health literacy among patients (Giebel et al., 2024).

Due to a lack of formal support from the state in many LMICs, caregivers play a vital role as informal care providers in mental health care, often acting as intermediaries between patients and providers (Daliri et al., 2024). There are evidences from research on commonly reported experiences of stress, tiredness and frustration associated with this role of caring for a mentally ill relative (Mulud and McCarthy, 2017). While patients in LMICs may use digital tools to support appointment management, medication adherence and health information sharing, caregivers' perspectives on mHealth tools remain insufficiently studied. Specifically, little is known about how caregivers in Nigeria perceive the relevance, usability and value of mobile health care within their caregiving responsibilities. Exploring these caregiver perspectives is essential for developing digital interventions that are both effective and contextually appropriate, with the potential to ultimately enhance mental health outcomes through improved support networks.

In Nigeria's under-resourced mental health landscape, mHealth holds significant promise to improve access to and continuity of services, but little is known about how key stakeholders interpret and use digital tools. In the context of facilitating the use of mobile health in mental health care, human-centred digital approaches, grounded in the perspectives of patients, caregivers and providers are critical for ensuring that mobile health tools are acceptable, feasible, culturally appropriate, and scalable.

This study aims to explore the perspectives of patients, caregivers and healthcare providers on the use of mobile health tools to facilitate access to and continuity of mental health care.

## Methods

### Participants and study setting

This qualitative study was conducted in Ibadan, Oyo State, in Southwest Nigeria, a region with a mix of urban, semi-urban and rural communities. The study is part of a larger project funded by the National Institute of Mental Health designed to develop measures of access to mental health services in Nigeria. We used Key Informant Interviews (KIIs) to gather information from the study participants recruited through purposive sampling from nine primary health clinics. Data were collected through Key Informant Interviews (KIIs) carried out across selected primary health clinics for their involvement in mental health service delivery. This study specifically focused on a subset of participants reporting experience using digital health tools for mental health care. All participants were aged 18 years or older and included: (1) patients with a diagnosis of depression, (2) caregivers of individuals with depression, and (3) healthcare providers involved in the care of such patients.

A total of 17 participants were interviewed, comprising persons with lived experience (PLE) of depression (n = 7), primary healthcare workers (n = 7), and caregivers (n = 3). Semi-structured interview questions were developed by JF, LK and TF to explore factors that impede or facilitate access to and continuity of mental health care. Examples of the semi-structured interview questions asked are shown in Table 1.

### Data collection

Primary care staff identified patients with documented depression through clinic record books and electronic records. Healthcare providers initially contacted patients to introduce the research and to seek their consent to share contact information with the research team. The study coordinator (TF) then contacted PLE by phone to arrange interviews at their preferred location, either at home or a private room

**Table 1.** Participants interview questions

| Participant groups | Questions |
| --- | --- |
| Patients | Was there anything that made it easier to start receiving mental health care? Was there anything that made it easier to stick with or keep going to seek mental health care? Of all the things that made it easier, which was the most helpful? |
| Healthcare providers | Was there anything that made it easier for patients to start receiving care? Was there anything that made it easier for patients to stick with or keep going to seek care? Of all the things that made it easier, which was the most helpful for patients? |
| Caregivers | Was there anything that made it easier for your relative to start receiving mental health care? Was there anything that made it easier for your relative to stick with or keep going to seek mental health care? Of all the things that made it easier for your relative, which was the most helpful? |

at the clinic, ensuring privacy. Caregivers of selected patients were invited to participate in the study where appropriate, based on the patient's recommendation. Their inclusion aimed to capture complementary insights into facilitators and barriers to mental health care from the perspective of those supporting patients through the treatment process. Primary care providers were also recruited to explore systemic and service-level challenges, drawing on their clinical experiences and delivery of mental health services. Verbal consent was obtained over the phone prior to scheduling the interviews. The research assistants, BO (MBBS) and TO (BA), both trained in qualitative research methods, provided detailed information about the study's purpose, procedures, potential risks and benefits to all participants before the interview and consent, with opportunities to ask questions to ensure informed consent. Written consent was documented for all participants. Confidentiality and privacy were rigorously maintained throughout recruitment and data collection. No non-participants were present during the interviews.

Each interview lasted approximately 45 minutes and was conducted in person by two female research assistants (BO and TO), who had no prior relationship with participants. They were introduced to the participants as members of the research team. Interviews were primarily conducted in Yoruba, the predominant local language. The interviewers documented brief observational field notes during each interview to capture participants' impressions. All Key Informant Interviews (KIIs) were audio-recorded, transcribed verbatim, translated into English for analysis using the World Health Organization (WHO) back translation methodology (World Health Organization, 2010), and analyzed shortly thereafter. Transcripts were not returned to participants for comments or corrections. Responses were de-identified, and transcripts were securely stored with access restricted to the core research team. All 17 KIIs were conducted over a five-month period in 2025, with no repeat interviews conducted.

## Data analysis

Transcriptions were coded by BO and TO, who are trained and experienced qualitative researchers. A qualitative thematic analysis approach was used. Data were analyzed using inductive thematic analysis following the six-phase approach described by Braun and Clarke (Braun and Clarke, 2006). BO and LK generated initial codes, which were then reviewed and agreed on with JF and TF. In the data analysis process, duplicate subcodes were merged and redundant codes were removed. Together, the team developed and refined the inductive coding framework. BO and LK independently coded all transcripts, identifying text relevant to key themes. Intercoder agreement was reached through discussion and consensus among team members to resolve discrepancies. Throughout the coding process, JF, LK and BO held regular meetings to discuss code definitions and maintain consistency. The team generated summaries indicating the frequency of each code, providing an overview of barriers and facilitators to access. All authors contributed to the interpretation of the study findings. Thematic analysis was conducted inductively using NVivo 15 software. Participants did not provide feedback on the findings. This study is reported in accordance with the Consolidated Criteria for Reporting Qualitative Research (COREQ) checklist.

## Results

### Demographics

The mean age of the patients was 47.9 (SD 20.3) years. They were predominantly female patients (6/7, 86%), and most were married

**Table 2.** Demographic information of participants

| Variables | Values | | |
| --- | --- | --- | --- |
| | Persons with Lived Experience of depression N(%) or μ(SD) | Caregivers N(%) or μ(SD) | Healthcare workers N(%) or μ(SD) |
| Sex | | | |
| Female | 6 (86) | 1 (33) | 6 (86) |
| Male | 1 (14) | 2 (67) | 1 (14) |
| Age in years, μ (SD) | 47.9 (20.3) | 52.7 (11.6) | 51 (4.1) |
| Level of education, n (%) | No formal education, 1 (14) Primary, 1 (14) Secondary, 2 (29) Tertiary, 3 (43) | Secondary, 1 (33) Tertiary, 2 (67) | – |
| Employment, n (%) | Unemployed, 2 (29) Self-employed, 4 (57) Retired, 1 (14) | Self-employed, 2 (67) Full-time employed, 1 (33) | – |
| Marital status, n (%) | Single, 1 (14) Married, 5 (71) Divorced, 1 (14) | – | – |
| Years of work experience in mental health, μ (SD) | – | – | 9.3 (4.8) |

(5/7, 71%). More than half of the patients were self-employed (4/7, 57%) with varied levels of education. The caregivers were middle aged (mean 52.7, SD 11.6), were predominantly male (2/3, 67%) had several years of formal education and a broad range of work experiences. The healthcare workers were mostly female (6/7, 86%) with a mean age of 51 (SD 4.08) years. They had an average of 9.3 (SD 4.8) years' work experience in mental health care. A detailed summary of participants' demographics is presented in Table 2.

### Engagement with mHealth for self-care

#### The perspectives of persons with lived experience (PLE)

PLE in this study highlighted the supportive role of audiovisual content delivered through a mobile app introduced to them at the clinic as part of their treatment package. The majority had a history of perinatal depression and described how access to treatment videos helped them with self-care and emotional regulation. These audiovisual materials were designed to provide both mood-lifting content and practical guidance for managing symptoms.

Two patients shared:

*"She gave me access to some engaging videos that I could have on my phone to watch and make me happy."*

*"She told me to watch education videos that could help me to take care of myself and stop thinking too much. I used to watch it and that helped my mood a lot."*

In addition to the educational and mood-enhancing content, patients also valued regular communication from healthcare workers through phone calls and text messages, which reinforced their sense of connection to care and encouraged treatment adherence.

Such reminders were critical in helping patients stay engaged and maintain continuity of care.

One participant noted, *"There was a healthcare worker that was always calling and sending text messages to remind me to come for my appointments."*

Another PLE echoed this experience: *"They call me often and they also send text messages to remind me about my appointments,"*

*"They usually call me to remind me to come to the clinic."*

Recognizing that not all patients own phones, some providers adapted by contacting family members to ensure messages reached patients.

As one participant shared, *"They do call me, I don't have a phone, but they call my mother to remind me."*

This approach highlights the flexible strategies used to overcome barriers in communication. Patients also described a sense of community and connection fostered by digital platforms.

One explained, *"They created a platform for us where patients can comment and we can know our next clinic date."*

Such platforms appear to enhance transparency and active participation in care management.

### Healthcare workers' perspective

Healthcare providers in this study described the use of mobile phones as a central part of their patient engagement and follow-up strategy. For many, regular phone calls and text messages served as vital tools for maintaining contact with patients, particularly in a context where inconsistent attendance at appointments can hinder treatment continuity.

One provider explained, *"We usually call them to remind them about their appointments,"* highlighting how such reminders are woven into routine care practices. These efforts were not just administrative; they reflected a deliberate strategy to reduce missed appointments.

Another provider added, *"We also call and send them text messages to remind them about their appointments."*

Beyond appointment reminders, Healthcare providers described a proactive and compassionate approach to patient follow-up, underscoring the importance of regular communication. Many reported calling their patients at least once a week to check in and offer ongoing support. These weekly calls serve multiple purposes: confirming upcoming appointments, monitoring patients' health status and providing reassurance. Additionally, providers emphasized the importance of checking on patients' well-being beyond appointment logistics. As one shared, *"We also call them to check on their health, if they are feeling better."*

Healthcare providers also described innovative approaches to patient education using mobile technology. One provider shared an example of a program where they sent educational videos to patients, explaining key information and then engaging patients by asking what they had learned. This interactive method allowed providers to identify misunderstandings and correct any incorrect answers in real time.

The provider reflected, *"There was a program we did where we sent them videos, explained to them and asked them what they learnt. We modify their wrong answers and I think that really helps them."*

### Caregivers perspectives

All the caregivers identified the benefit of being able to contact the care provider on the phone. One caregiver described how having a direct line of communication with a healthcare worker transformed their experience of navigating mental health care. Before this

**Table 3.** Key themes and perspectives of participants on mHealth use in mental health care

| Characteristics | Key themes | Perspectives |
| --- | --- | --- |
| Patients | Educational Audio-visuals | • Mood-lifting<br>• Practical self-care. |
|  | Phone calls and SMS | · Clinic appointment reminders<br>· Treatment adherence |
| Caregivers | Phone calls and SMS | • Connection to the health system<br>• Supported uninterrupted care<br>• Alleviate feelings of isolation |
| Healthcare Providers | Educational Audio-visuals | • Care management. |
|  | Phone calls and SMS | · Clinic appointments reminders and follow-up. |

connection, they often felt lost in the system, unsure of when to bring their relative in for follow-up or how to interpret symptoms that arose between visits. But once they were given the phone number of a familiar healthcare provider, things changed.

*"Now that we have the phone number of the health workers there, it made it easy for us. All I have to do is text her and she will tell us what to do and when to come."*

For many caregivers, one of the most reassuring aspects of the mental health care experience was the consistent and compassionate follow-up by healthcare workers. These weren't just routine check-ins, but they were messages that signaled presence, care and continuity. Caregivers described how regular texts or calls helped bridge the often-fragmented nature of care and provided a vital sense of connection to the health system.

*"They kept checking on my daughter to be sure she was fine. They would send messages to ask about her welfare. It went a long way in the treatment that improved her condition."*

These messages went beyond logistical reminders. They conveyed that someone was paying attention, that their loved one's well-being mattered even outside the walls of the clinic. For caregivers who often carried the emotional and physical burden of supporting patients with depression, this kind of ongoing engagement helped alleviate feelings of isolation and uncertainty.

*"The one that was the most helpful was that they did not forget us at all, at any time. They were always checking up on us."*

Caregivers also noted how healthcare providers went beyond standard practices to maintain continuity of care, particularly when appointments were missed. Instead of passively waiting for patients to return, providers actively reached out to check on their well-being. These follow-ups served as practical ways of uninterrupted care (see Table 3).

*"The days we do not come even after the reminders they send, they always call us to ask about my daughter's whereabouts. They will want to know what happened to her that prevented her from the clinic visit. They were caring."*

### Discussion

The perspectives of PLE, healthcare providers and caregivers underscore the promise of mHealth as a scalable, contextually appropriate strategy to enhance mental health service accessibility, continuity

and quality within an under-resourced health system marked by critical workforce shortages and infrastructural limitations.

From the perspective of PLEs, the provision of mobile phones with engaging and educational applications highlights the potential of digital tools to extend psychosocial support beyond in-person clinical encounters. This finding resonates with prior evidence demonstrating that multimedia mHealth interventions improve emotional well-being and promote adaptive coping in mental health populations (Mens et al., 2022; Diano et al., 2023; Willis and Neblett, 2023). Patients' appreciation for regular calls and SMS reminders further confirms the utility of simple yet effective strategies to mitigate forgetfulness and logistical barriers, well-documented challenges contributing to high rates of missed appointments in LMIC settings (Opon et al., 2020; Brancewicz et al., 2025). Importantly, the dual use of calls and texts addresses varied patient preferences and technology access, reflecting recommendations for culturally sensitive and patient-centered mHealth implementation (Opon et al., 2020). The strategy of contacting family members when patients lack personal phones demonstrates adaptive solutions to digital inequities that could otherwise exclude the most vulnerable from care (Glenton et al., 2024). The creation of digital platforms facilitating patient interaction and clinic scheduling not only enhances transparency, which is a key factor in reducing stigma and empowering patients within marginalized mental health communities (Naslund et al., 2016; Simbo, 2024).

The perspectives of healthcare providers reinforce the critical role of mobile phones as tools for proactive engagement in mental health care delivery. Providers' routine use of appointment reminders through calls and texts aligns with evidence showing that such reminders significantly improve treatment adherence and reduce loss to follow-up in resource-constrained environments (World Health Organization, 2019; Opon et al., 2020). The use of both voice calls and SMS as communication channels reflects approaches recommended in the literature to accommodate diverse patient preferences and needs. Voice calls can facilitate more personalized interaction, while SMS offers a discreet way to deliver reminders, which may help address stigma-related concerns and improve message reach despite privacy and technological constraints (Curioso et al., 2009; Materia et al., 2023). Follow-up calls exemplify a holistic care approach, extending support beyond appointment logistics to include health monitoring and emotional reassurance, consistent with literature emphasizing relational continuity as foundational for sustained engagement in mental health treatment (Kwobah et al., 2021). Moreover, the innovative use of educational videos with interactive feedback mechanisms illustrates how mHealth can enhance health literacy and patient empowerment, critical determinants of adherence and clinical outcomes in mental health care (Yueh-Hsiu and Meei-Fang, 2021; Deniz-Garcia et al., 2023).

Caregiver perspectives highlight how mHealth tools enhance caregivers' roles by improving communication with providers, easing caregiving burden and fostering trust in the mental health system. Caregivers' experiences of direct contact with healthcare workers and regular welfare checks through mobile calls mirror findings from other LMIC contexts where mobile technologies strengthen caregiver support and involvement in mental health care (Ferré-Grau et al., 2021; Perez et al., 2022; Turnbull et al., 2024; Shen et al., 2025). These supportive interactions go beyond administrative functions to serve as expressions of empathy, which is particularly crucial in settings where stigma and resource constraints intensify caregiver stress (Mei et al., 2025). The proactive outreach to patients who miss appointments reflects a commitment to family-centered care models, emphasizing continuity and relapse prevention through accessible digital means. Including caregivers' perspectives was a key strength of this study, providing insights into support systems and engagement with mental health interventions. This approach equally raises confidentiality concerns, as involving caregivers in someone else's care may risk disclosure of confidential information. We reduced this by obtaining informed consent from both patients and caregivers and clearly defining the scope of information shared. Future interventions should continue to balance the benefits of caregiver engagement with the need to protect patient privacy.

The perspectives on mHealth use in mental health care differ across patients, caregivers and healthcare providers, reflecting their unique roles and needs. Patients value mHealth primarily for its role in not forgetting appointments, mood uplift and practical self-care, with tools like phone calls, SMS and educational audio-visuals enhancing their engagement and understanding. In contrast, caregivers emphasize how mHealth helps alleviate feelings of isolation, strengthens connection to the health system and ensures uninterrupted care, while also aiding in depression management. Meanwhile, healthcare providers focus on mHealth's ability to streamline clinic appointment reminders, enable patient follow-up and support care participation, highlighting its utility in maintaining continuity and efficiency of care. While all groups recognize the value of phone and multimedia tools, their priorities reflect their distinct positions in the care ecosystem.

## Implications for practice and research

Collectively, these findings reinforce the WHO's recommendations advocating for the rapid expansion of digital health solutions to address mental health service gaps in LMICs. mHealth tools, from basic SMS reminders to multimedia educational content and interactive platforms, hold potential to support mental health care delivery in Nigeria. They help overcome barriers such as limited access to mental health education, forgetfulness around clinic appointments and lack of timely follow-up. Importantly, the integration of mobile technologies with empathetic, relationship-centered care highlights that successful mHealth interventions depend not only on technological functionality but also on maintaining human connection and trust.

Although the shortage of trained mental health professionals is a barrier in LMICs, health care providers in this study were able to integrate calls and SMS reminders into routine follow-up procedures. While resource constraints remain a challenge, this approach shows that simple adaptations can allow providers to offer mHealth support without substantially increasing workload. Future interventions should consider strategies such as task-shifting with trained lay health workers or automated messaging systems to enhance scalability and sustainability in resource-limited settings.

Future research should rigorously evaluate the impact of these mHealth strategies on clinical outcomes and cost-effectiveness within Nigerian and similar LMIC settings, while exploring ways to scale interventions sustainably and equitably. Given the limitations in universal digital access, integrating mHealth with traditional face-to-face services through a hybrid care approach is essential to ensure inclusive mental health support without exacerbating access disparities.

In conclusion, this study underscores the multifaceted value of mHealth as a feasible, acceptable and impactful approach to enhancing mental health care access, continuity and quality in Nigeria's resource-limited settings. By centering the voices of key stakeholders,

these insights offer critical guidance for designing and implementing digital health interventions that are responsive to local realities and poised to narrow the substantial mental health treatment gap.

## Limitations

This study has several limitations. First, the interview guide was not primarily designed to ask about mHealth and our analysis focused on the sub-sample of participants who mentioned it. Consequently, participants mostly reported on facilitators to the use of mHealth tools. Although participants did not explicitly report barriers to mHealth use, several potential challenges are likely in low-resource contexts like Nigeria. Prior research highlights barriers that could affect uptake or engagement with mHealth interventions, such as limited access to smartphones, unreliable internet or mobile networks, digital literacy issues and other demands competing with patients' time.

Second, the findings are drawn from qualitative data collected within a specific geographic and cultural context in Nigeria, which may limit generalizability to other regions or LMICs. Third, the reliance on self-reported data introduces the possibility of social desirability bias, potentially influencing participants to present more favorable views of mHealth use. Finally, the study did not include quantitative measures of clinical outcomes, restricting the ability to assess the direct impact of mHealth on mental health status. Future research should address these limitations by incorporating broader and more diverse samples, including data on barriers to mHealth use to inform the design of scalable and appropriate mHealth strategies and utilizing mixed-methods approaches to better understand the effectiveness and accessibility of mHealth interventions.

**Open peer review.** To view the open peer review materials for this article, please visit http://doi.org/10.1017/gmh.2026.10137.

**Data availability statement.** The data that support the findings of this study are available from the corresponding author, BO, upon reasonable request.

**Authors contribution.** All authors made substantial contributions to either the conception or design of the work or the acquisition, analysis, or interpretation of data. JF,LK,TF designed the semi-structured interview questions. BO,LK,JF interpreted and analysed the data. All authors drafted or revised the manuscript critically for important intellectual content and approved the final version to be published. All authors agree to be accountable for all aspects of the work and ensure that questions related to the accuracy or integrity of any part of the work are appropriately investigated and resolved.

**Financial support.** This work received no dedicated funding and was developed out of a grant from the National Institute of Mental Health (R34MH137174) awarded to Drs Fortney and Kola.

**Competing interests.** The authors declare none.

**Ethics statement.** The study received ethical approval from the University College Hospital/University of Ibadan and University of Washington Institutional Review Boards, in accordance with the 1964 Declaration of Helsinki and its subsequent amendments. Consent forms were securely stored in the research file. Participants were free to decline answering any questions during the interviews. Participants were compensated $10 for completing the interview.

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
