## [Reviewer Report]

Dear Authors,

Thank you for the chance to review this manuscript. It delivers a relevant and insightful investigation of mobile health use in Nigeria’s mental health landscape, thoughtfully integrating patients‘, caregivers’, and providers’ perspectives. I thank the authors for their efforts and encourage them to keep expanding this line of inquiry to help ensure LMICs' voices are meaningfully represented in digital mental health research.

Introduction

1. The Gureje et al. (2006) citation is helpful, though it seems a bit outdated. Could the authors consider citing more recent data?

2. Bringing caregivers’ perspectives is the study’s most notable contribution. Their value is clearly explained in the introduction on page 4, lines 43-52. I also think the value of human-centred digital solutions (as stated in the impact statement) is another noteworthy contribution. I would suggest providing context for this in the introduction.

Methods

3. Lines 21 – 22. It appears there might be a minor grammar issue. Please revise it.

4. Lines 32-33 ‘Examples of semi-structured interviews are:’. Did you mean examples of semi-structured interview questions?

5. “Confidentiality and privacy were rigorously maintained throughout recruitment and data.” Did you mean throughout recruitment and data collection?

6. For clarity and transparency, could the authors specify the thematic analysis guideline used and cite the relevant source?

Results

7. I understand that the results may be concise due to word count restrictions or because the excerpts originate from an interview guide not primarily focused on m-Health for mental health. That said, including a bit more detail or context where possible could enhance the reader’s understanding.

8. Regarding the educational audio-visual materials, could the authors clarify whether the health providers designed these? Additionally, where were these resources sourced from, and were all participants referring to the same content or different materials?

Discussion

9. Koshy et al., (2008), reference is helpful but might be outdated. Can you find a more recent reference?

10. As mentioned previously, bringing the caregiver’s perspectives is the study’s notable contribution. However, it is important for the authors to also reflect on the confidentiality implications that may result from this approach, i.e., calling the caregiver and involving them in someone else’s care. I think both the positive and negative aspects of this approach can be acknowledged.

11. In the introduction, the authors make an important point about the shortage of trained mental health professionals, a challenge faced by many LMICs. To strengthen the discussion, could the authors elaborate on how the health providers managed to find time for calls and sending SMS reminders to patients and caregivers? Were there enough resources? This is important for the future development of such strategies.

12. Again, I understand that the interview guide was not primarily designed to ask about mHealth (thank you for acknowledging this limitation). If no barriers were mentioned by participants, given Nigeria’s context and previous m-Health research, can the authors reflect on potential barriers?

This is a relevant study that adds to the understanding of mental health challenges in LMICs and how we can leverage digital solutions. My comments are intended to strengthen clarity and provide additional context where possible. I hope these suggestions are helpful.

---

## [Reviewer Report]

Thank you for this important paper on the use of Digital Health/mHealth to reach those in low resource and difficult to reach settings. Your highlighting the use of text messages (SMS) to provide constant and compassionate follow up care is an important point. Your paper provides hope to those in low resource settings who are often without care, especially mental health care. Great work!

---

## [Reviewer Report]

Introduction -

- strong intro section as to why mHealth is important and why identifying perspectives on mHealth is important. To further strengthen the background, I would suggest adding in additional information (including any citations if they exist) on the role of caregivers as informal providers. This is an interesting topic and lines 40 - 52, page 4, can be a separate paragraph sharing this information.

- Because the aim of the paper is on exploring facilitation of access to and continuity of MH care, it would be helpful to clearly provide examples of this in the intro. I see that the paragraph starting with “While emerging evidence...” is aimed to provide this information, I would suggest more clearly identifying examples of how mobile health tools can facilitate access to and continuity of MH care in LMICs.

Methods -

- page 5, line 22 - across selected ___? seems to be missing a word

- page 5, line 33 - should it be interview questions?

- It would be helpful to add a table of questions that were asked of the various participant groups in the study.

- For participant selection, how many people were contacted to take part in the qualitative interviews? Did anyone refuse?

- Were field notes taken during data collection? How was data saturation established?/how was the sample size determined?

- Could you share details about the coding tree? How many codes were developed?

- I would suggest following a qualitative reporting checklist such as the COREQ. Noting that you have used such a checklist in them methods section and making sure all sections of the checklist are covered in the paper would make the methods section stronger.

Results

- The figure is helpful but rather than just key words such as “treatment adherence” it would be helpful if this figure could summarize the key perspectives of the three groups.

- A table of the key themes and perspectives would be helpful or the figure could be changed to better address this. Additional quotes from the interviews could also be added to this.

Discussion section is strong and I appreciate the addition of implications for practice.

---

## [Editor Report]

Dear Authors 

We have received reviewer comments on your manuscript. In light of that, we recommend minor revisions for us to consider your manuscript for publication. 

Regards

Siham

---

## [Editor Report]

Dear Authors

Thanks for sending in the revisions to your manuscript in light of the reviewer comments. 

We will inform you of the next steps for its publication. 

Regards

Siham